# The 1B vaccine strain of *Chlamydia abortus* produces placental pathology indistinguishable from a wild type infection

Sergio Gaston Caspe[1,2,3]*, Morag Livingstone[1], David Frew[1], Kevin Aitchison[1], Sean Ranjan Wattegedera[1], Gary Entrican[1], Javier Palarea-Albaladejo[4], Tom Nathan McNeilly[1], Elspeth Milne[2], Neil Donald Sargison[2], Francesca Chianini[1], David Longbottom[1]

**1** Moredun Research Institute, Penicuik, Midlothian, United Kingdom, **2** Royal (Dick) School of Veterinary Studies, University of Edinburgh, Edinburgh, United Kingdom, **3** Estación Experimental Mercedes, Instituto Nacional de Tecnología Agropecuaria (INTA), Mercedes, Corrientes, Argentina, **4** Biomathematics and Statistics Scotland, Edinburgh, United Kingdom

\* gaston.caspe@moredun.ac.uk

## Abstract

*Chlamydia abortus* is one of the most commonly diagnosed causes of infectious abortion in small ruminants worldwide. Control of the disease (Enzootic Abortion of Ewes or EAE) is achieved using the commercial live, attenuated *C. abortus* 1B vaccine strain, which can be distinguished from virulent wild-type (wt) strains by polymerase chain reaction-restriction fragment length polymorphism (PCR-RFLP) analysis. Published studies applying this typing method and whole-genome sequence analyses to cases of EAE in vaccinated and non-vaccinated animals have provided strong evidence that the 1B strain is not attenuated and can infect the placenta causing disease in some ewes. Therefore, the objective of this study was to characterise the lesions found in the placentas of ewes vaccinated with the 1B strain and to compare these to those resulting from a wt infection. A *C. abortus*-free flock of multiparous adult ewes was vaccinated twice, over three breeding seasons, each before mating, with the commercial *C. abortus* 1B vaccine strain (Cevac® Chlamydia, Ceva Animal Health Ltd.). In the second lambing season following vaccination, placentas (n = 117) were collected at parturition and analysed by *C. abortus*-specific real-time quantitative PCR (qPCR). Two placentas, from a single ewe, which gave birth to live twin lambs, were found to be positive by qPCR and viable organisms were recovered and identified as vaccine type (vt) by PCR-RFLP, with no evidence of any wt strain being present. All cotyledons from the vt-infected placentas were analysed by histopathology and immunohistochemistry and compared to those from wt-infected placentas. Both vt-infected placentas showed lesions typical of those found in a wt infection in terms of their severity, distribution, and associated intensity of antigen labelling. These results conclusively demonstrate that the 1B strain can infect the placenta, producing typical EAE placental lesions that are indistinguishable from those found in wt infected animals.

**Data Availability Statement:** All relevant data are within the manuscript and its Supporting Information files.

**Funding:** ML, DF, KA, SRW, JPA, TNM, FC and DL are supported by the Scottish Government Rural and Environment Science and Analytical Services Division (RESAS) Strategic Research Programme, 2016–2021. SGC is supported by Instituto Nacional de Tecnologia Agropecuaria (INTA), Argentina. The funders had no role in study design, data collection and analysis, decision to publish, or preparation of the manuscript.

**Competing interests:** The authors have declared that no competing interests exist.

## Introduction

*Chlamydia abortus*, an obligate intracellular Gram-negative bacterium, is the etiological agent of an abortifacient disease variously known as Enzootic Abortion of Ewes (EAE), Ovine Enzootic Abortion (OEA), ovine chlamydiosis or chlamydial abortion [1]. EAE is one of the most commonly diagnosed causes of infectious abortion in small ruminants around the world, with the exception of Australia and New Zealand [2,3]. In the UK, around 8.6% of flocks have been estimated to be affected, with an associated annual economic impact of upwards of £20M [4], while a field study has estimated the costs associated with chlamydial abortion storms to be in the region of £2,163 per 100 ewes [5].

The disease is characterised by late-term abortions (generally occurring in the last 2–3 weeks before expected lambing), stillbirths, or the delivery of weak lambs that fail to survive beyond 48 hrs [6,7]. The most significant sources of infection for transmission of the disease are the products of abortion, in particular, the heavily infected placentas, vaginal fluids, and the coats of dead fetuses or live lambs born from infected ewes. Bacterial loads present in or on these products of abortion/lambing are extremely high, reaching greater than $10^7$ chlamydial genomes per μl of extracted material [7]. These products can contaminate the bedding and pasture with *C.abortus*, facilitating the transmission of infectious organisms to naïve animals through inhalation and ingestion [6].

The severe mixed infiltration of inflammatory cells, interstitial oedema and necrosis of the chorionic epithelium (trophoblast cells) during chlamydial infection affect the efficiency of the exchange of oxygenated blood and nutrients at the maternal-fetal interface, as well as the placental production of progesterone, which is responsible for the maintenance of pregnancy [8]. In addition, the ensuing vascular damage and thrombosis can cause ischaemic necrosis resulting from blockage of the oxygenated blood supply [9,10]. It is likely that a combination of all these factors results in the premature delivery of a dead fetus or weak lambs.

Control of infection can be achieved using the live attenuated temperature-sensitive *C. abortus* strain known as 1B, which has been successfully used in vaccines in many European countries [3,11]. This mutant strain was derived from the virulent field strain AB7 by chemical mutagenesis [12]. However, a specific polymerase chain reaction-restriction fragment length polymorphism (PCR-RFLP) typing assay that specifically differentiates the 1B vaccine strain from a wild-type (wt) strain has been developed and used to provide evidence suggesting that the 1B strain has a role in causing disease [11,13,14]. A recent analysis of the genome of the 1B vaccine strain and another temperature-sensitive mutant strain (*C. abortus* strain 1H) that had reverted to virulence, revealed that they are genetically identical, suggesting that there was no genetic basis for any attenuation in the 1B strain [3]. Furthermore, the 1B vaccine strain has very recently been reported to have been recovered from unvaccinated animals that had been in contact with vaccinated animals and which had aborted as a result of EAE, suggesting that the 1B vaccine strain can be naturally-transmitted [3,11]. However, there have been no studies investigating placental pathology resulting from infection with the vaccine strain. Therefore, the aim of this study was to characterise the lesions produced in the placentas of ewes vaccinated with the *C. abortus* 1B vaccine strain and compare them with those observed during a wt infection.

## Materials and methods

### Ethics statement

Placentas used in the present study to test for the presence of *C. abortus* 1B strain were collected from a flock of Texel-cross ewes that had been vaccinated with the commercial *C.*

*abortus* 1B live, attenuated vaccine (see next section) on a working farm at parturition. These ewes had previously been purchased at a market from a Premium Health Scheme-accredited *C. abortus*-free flock. As no experimental procedures, sampling or any other interventions were conducted on these animals, and placentas only were collected as part of normal lambing, this part of the study was not subject to any Home Office licensing requirements.

Positive control formalin-fixed placental samples (wt-P1 and -P2) were used in this study for comparative purposes. These samples originated from a previously conducted study, from two ewes that had been experimentally inoculated with *C. abortus* strain S26/3 and both of which had aborted, one at 130 and the other at 133 days of gestation [15]. Negative control placental samples (Neg-P1 and -P2) also originating from this same study were taken from EAE-negative ewes. The study described here was carried out in strict accordance with the Animals (Scientific Procedures) Act 1986 and in compliance with all UK Home Office Inspectorate regulations. The experimental protocol was approved by the Moredun Experiments and Ethical Review Committee (Permit number: E30/17). All animals were monitored throughout the study for any clinical signs of ill health at least three times daily and any findings were recorded. Any animal found to be suffering or requiring treatment, for example from bacterial infections, was given appropriate veterinary care (including use of antibiotics by a registered veterinary practitioner) in accordance with standard veterinary practice.

### Collection of placentas and initial bacteriological screening

Placentas (n = 117) were collected from 75 Texel-cross ewes (from the same flock) at parturition. These ewes had been vaccinated on-farm as part of standard farming practice with a commercial 1B vaccine (Cevac® Chlamydia, Ceva Animal Health Ltd.) administered via the subcutaneous route according to the manufacturer's instructions. The ewes had been vaccinated twice (n = 16 vaccinated in 2015 and 2017; n = 59 vaccinated in 2016 and 2017), each approximately five weeks prior to mating. There were no reports of abortion occurring in this flock and all ewes lambed 'apparently healthy' live lambs. Placentas were collected at lambing, and each placenta was bagged and labelled immediately following delivery and transported directly to Moredun Research Institute for analysis. All placentas were assessed macroscopically for lesions and the percentage of the surface affected was estimated as described previously [16,17]. Smears from placentas showing lesions consistent with EAE were stained by modified Ziehl Neelsen (mZN) to confirm the presence of chlamydial organisms [18,19]. Samples of cotyledons and surrounding intercotyledonary membrane were excised from all placentas for the recovery and isolation of live *C. abortus* organisms, and for molecular and pathological analyses, as detailed in the next sections, ensuring that any areas exhibiting gross pathology were sampled.

### Placenta sampling and sensitivity of detection methods

As it was possible for only one cotyledon to be positive for *C. abortus* in a pool of cotyledons, it was necessary to determine the sensitivity of the *C. abortus* quantitative real-time PCR (qPCR) (described below). This was achieved in a series of spiking experiments testing different proportions of a single *C. abortus* positive cotyledon ($6x10^6$ *C. abortus* genome copies per μL of total extracted DNA) mixed with an increasing numbers of negative cotyledons from *C. abortus*-free placentas by qPCR following extraction of DNA from the pooled tissue. Ratios of positive: negative cotyledons ranging from 1:1 to 1:15 were tested. Using this methodology, it was determined that a single positive cotyledon could be detected from a pool of up to 15 cotyledons with 95% confidence [20].

Hence, one half of fifteen selected cotyledons was excised from each of the collected placentas and pooled for molecular analyses (see below). The remaining halves of each cotyledon were left *in situ* and the placentas were fixed in 10% neutral buffered formalin for haematoxylin and eosin staining (HE) and immunohistochemical studies (IHC).

Each pool of half-cotyledons (n = 15) from each of the 117 placentas was finely chopped using new sterile disposable scalpels and forceps for each placenta. The resulting mix was homogenised for 30 sec in a stomacher bag (Stomacher 80® Biomaster, Seward Limited, West Sussex, UK) at medium speed (265 rpm). Each mix was aliquoted into sterile 2 ml tubes and stored at -20°C for qPCR and PCR-RFLP analyses and possible future bacteriological isolation.

## Bacteriological isolation

Isolation of *C. abortus* was performed in HEp-2 (Human Epithelial type 2) cells as follows. A pooled sample of placental cotyledons was ground up in 4 ml sucrose–phosphate–glutamate transport medium [21] using a mortar and pestle and sterile sand then centrifuged at 100 x *g*. The supernatant was collected before consecutive passages through 0.8 and 0.45 μm filters (Sterile Acrodisc® Syringe Filters, Ann Arbor, USA). After filtration, the sample was inoculated onto HEp-2 cells grown on coverslips in Trac bottles (Thermo Scientific, Newport, UK) in Iscove's Modified Dulbecco's Medium (IMDM, Thermo Fisher Scientific, New York, USA), supplemented with 2% heat-inactivated foetal calf serum (Merck Life Science UK Limited, Gillingham, UK), 50 μg/mL gentamicin, 200 μg/mL streptomycin, 25U/mL nystatin and 1 μg/mL cycloheximide (Merck Ltd., Poole, UK). Trac bottles were centrifuged at 3,000 × *g* at room temperature for 2 × 15 min and incubated at 37°C in 5% $CO_2$ for 2 h before replenishing with fresh medium and then maintained under the same conditions. After 72 h, coverslips were fixed in methanol, stained using Giemsa 'Gurr' (Merck Ltd., Poole, UK), and examined for the presence of chlamydial inclusions by light microscopy.

## Quantitative real-time PCR and RFLP analyses

Total DNA was extracted from isolated organisms (see Bacteriological isolation) and representative 20 mg samples of pooled placental material (n = 15 cotyledons) using a DNeasy Blood & Tissue kit (Qiagen Ltd., Crawley, UK), according to the manufacturer's instructions. DNA samples were eluted in 200 μl of supplied buffer AE and stored at -20°C for molecular analyses.

Quantitative real-time PCR was carried out on all extracted placental DNA samples using primers based on the outer membrane protein gene OmpA, as described previously [7]. Briefly, the PCR reaction consisted of 2X TaqMan® universal PCR master mix (Applied Biosystems, Warrington, UK), OmpA forward primer (5'-CGGCATTCAACCTCGTT-3') and reverse primer (5'-CCTTGAGTGATGCCTACATTGG-3'), fluorescent probe (TaqMan® probe, 5' FAM-GTTAAAGGATCCTCCATAGCAGCTGATCAG-TAMRA 3'), 1 μL of DNA, and sterile Nuclease-Free water (Promega Corporation, Madison, USA) up to a final volume of 25 μL per sample. The thermal cycling conditions were 50°C for 2 minutes; 95°C for 10 minutes; 45 cycles of 95°C for 15 seconds; and finally, 60°C for 1 minute. Amplification and detection were performed using a QuantStudio 5 real-time PCR System (Applied Biosystems, Foster City, USA), following the manufacturer's standard protocols. Each sample was tested in triplicate and quantified against a standard curve (established with 10-fold concentrations ranging from $10^7$ to $10^1$ genome copies of *C. abortus* strain S26/3 per reaction). Concentrations of placental DNA were determined using a NanoDrop ND-100 (NanoDrop Technologies, Wilmington, USA) spectrophotometer, as previously published [7].

**Table 1. PCR-RFLP primers and fragment sizes for differentiation of *C. abortus* 1B vaccine strain from wild-type strains.**

| CDS[a] | Forward primer (5'-3') | Reverse primer (5'-3') | Amplicon size (bp) | Restriction endonuclease site | Restriction fragment sizes of PCR products (bp) | |
|---|---|---|---|---|---|---|
| | | | | | 1B strain | Wild-type strain |
| CAB153 | ATTAAAAGTAAGTGGAAAGATTTTACAACCTTA | TGAGGTCATACTTCTCTGTTTTGATTTTAT | 319 | *Sfc*I | 319 | 177+142 |
| CAB636 | AGTTTGTACTTTGATGAGAATTCCAATG | TGAGGTCATACTTCTCTGTTTTGATTTTAT | 118 | *Hae*III | 118 | 74+44 |
| CAB648 | TCACAAAAGCGATGCCCATC | TCTAAATCTCCGCATTCGG | 177 | *Sau*3AI | 88+89 | 67+21+89 |

[a] Gene IDs for *C. abortus* strain S26/3 (accession number CR848038).

All qPCR-positive samples were analysed further by PCR-RFLP to confirm if the samples were of a vaccine type (vt), wt or mixed infection origin. PCR-RFLP analysis differentiates vt and wt strains of *C. abortus* based on mutations in restriction enzyme sites in three genes (*Sfc*I in CAB153, *Hae*III in CAB636 and *Sau*3AI in CAB648 in *C. abortus* strain S26/3, genome accession number CR848038) [14,22] that prevent cleavage of PCR fragments derived from vt strains. The PCR primers used are detailed in Table 1, and all assay conditions were as previously published [14]. In addition to using genomic DNA prepared from the *C. abortus* 1B vaccine strain as a control, we also used the wild-type parental vaccine strain AB7 and UK wild-type strain S26/3 for comparison.

## Histopathological examination

All of the cotyledons from qPCR and PCR-RFLP positive placentas were fixed in 10% buffered formalin, trimmed, dehydrated through graded alcohol and xylene, and embedded in paraffin wax. Sections were cut at a thickness of 4μm, stained with HE and examined histologically.

The lesions observed in vt placentas were compared to those obtained from typical wt-EAE infected placentas. The latter were obtained from two ewes experimentally infected with *C. abortus* strain S26/3 (wt-placenta 1 (P1) cotyledons, n = 33; and wt-P2 cotyledons, n = 38) [15]. Placentas from two EAE-free ewes that lambed normally served as negative controls (Neg-P1 cotyledons, n = 40 and Neg-P2 cotyledons, n = 36).

## Immunohistochemistry

Formalin-fixed paraffin-wax embedded cotyledons were sectioned at 5 μm thickness, mounted on Superfrost® slides (Menzel-Gläser, Braunschweig, Germany) and labelled with a mouse monoclonal antibody (mAb) raised against the lipopolysaccharide (LPS) of *C. abortus* S26/3 strain (mAb clone 13/4, Santa Cruz Biotechnology, Heidelberg, Germany), as previously described [23]. Briefly, after dewaxing in xylene and rehydration in successive passages in alcohol and water, the slides were treated with 3% hydrogen peroxide (Merck Life Science UK Limited, Saint Louis, Missouri, USA) in methanol (v/v) to inactivate any endogenous peroxidase activity. After washing with tris-buffered saline (TBS), non-specific binding was blocked with 20% normal goat serum in TBS for 30 min. After overnight incubation with primary anti-chlamydial LPS mAb (diluted 1/500 in TBS), the slides were washed twice with TBS. Next, goat anti-mouse IgG conjugate (Envision™+ System HRP labelled polymer, Dako, Ely, UK) 1/20 (v/v) was applied for 30 min, according to manufacturer's instructions. After washing with TBS, the slides were incubated with aminoethyl carbazole alcohol soluble chromogen (AEC, Vector Laboratories, Peterborough, UK) for 8 min before washing in tap water and counterstaining with haematoxylin and mounted with mounting solution (Histomount™, Thermo

Fisher Scientific, Camarillo, CA, USA). Negative control sections were prepared for each sample substituting the primary mAb 13/4 with IgG1 isotype control from murine myeloma clone MOPC 21 (Merck Life Science UK Limited, Saint Louis, Missouri, USA) diluted 1/250 in TBS. A positive control section of placental tissue from a known ovine chlamydial abortion case was included in every run.

Slides were examined and scored, with grading based on intensity and extent of immunolabelling by immunohistochemistry in the trophoblast layer and mesenchyme (IHC) or in blood vessels (VIHC), from 0 to 5 (0, no labelling; 1, focal; 2, 2 to 4 foci; 3, multifocal; 4, locally extensive or multifocal to diffuse/intense; 5, diffuse/ very intense).

## Results

### Gross pathology, isolation and qPCR analysis

Two of the 117 placentas examined showed macroscopic lesions consistent with *C. abortus* infection (Fig 1 and S1 Fig). Both placentas (vt-P1 and -P2) were positive by qPCR ($5.4 \times 10^6$ and $2.6 \times 10^5$ genome copies per μL of total extracted DNA, respectively) and originated from

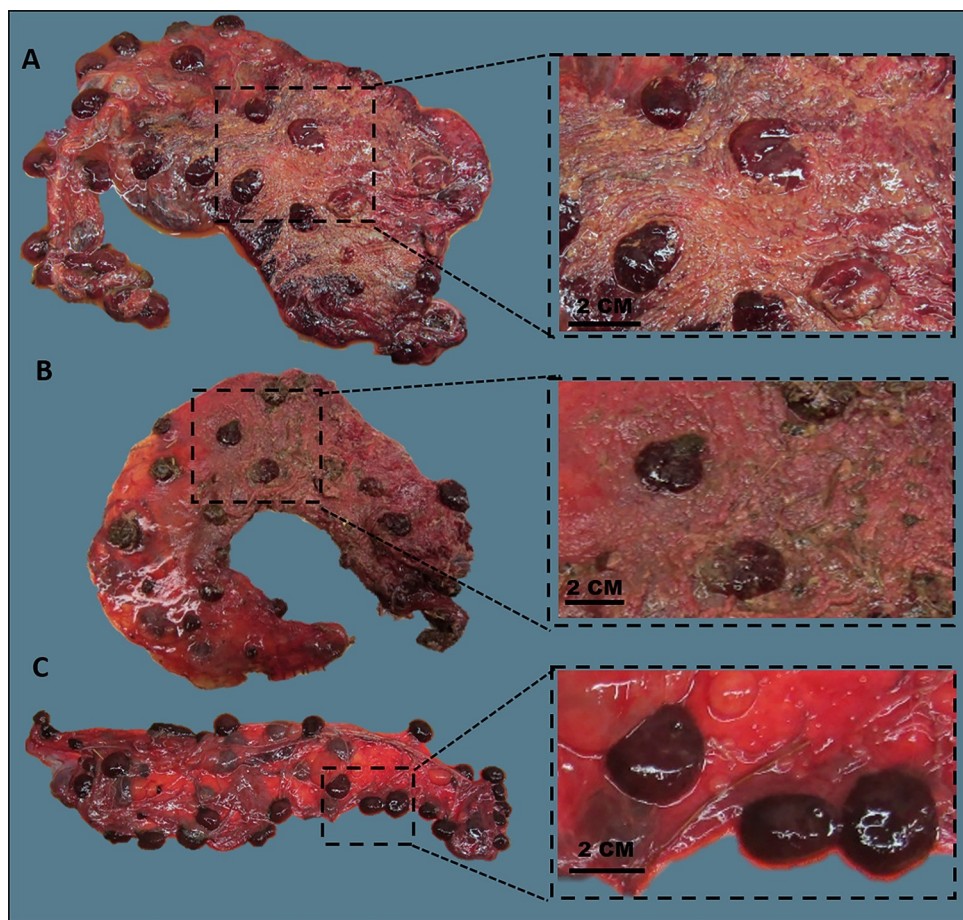

**Fig 1. Placentas infected with vaccine-type and wild-type *C. abortus* strains, and negative control.** (A) Placenta vt-P1: showing the oedema and thickening of the whole placenta covering and the typical dark red or grey-whitish colouration of the cotyledons and partially covered with a cream-coloured exudate on the intercotyledonary areas. (B) Placenta wt-P1: showing the oedema and thickening of the whole placenta partially covered with a cream-coloured exudate on the intercotyledonary area and dark red and grey cotyledons. (C) Placenta Neg-P1: showing red colour of the cotyledons, and thin and clear and translucent intercotyledonary membranes.

the same ewe (ewe number 6093), which lambed normally producing two live lambs. Live organisms were successfully isolated from these two placentas and both isolates were confirmed as qPCR positive for *C. abortus*. There were no gross lesions evident on any of the remaining 115 placentas, all of which were negative by qPCR (median and interquartile range (IQR) values equal to 30.79 and 99.07 genome copies per µL of total extracted DNA respectively; mean ± SD = 3.61 ± 1.4 in natural logarithm scale). These results are graphically summarized in S2 Fig.

Approximately 60% of vt-P1 showed lesions consistent with *C. abortus* infection. These changes were characterised by dark red or grey-whitish colouration of the cotyledons with an overlying creamy friable exudate that extended to the adjacent intercotyledonary areas. The intercotyledonary regions showed a thickened, oedematous appearance that is characteristic of EAE (Fig 1A). No gross lesions were evident on vt-P2 at the time of assessment and it was only subsequently that we noted vt-P2 was also positive for *C. abortus* following mZN and PCR analysis. After formalin fixation, we observed that approximately 5–10% of the surface area showed differences in colouration of the cotyledons, being paler than the rest of the cotyledons, and the intercotyledonary tissue in this area was slightly thickened (S1 Fig). Both placentas were positive by mZN and by qPCR, while bacterial loads in vt-P1 and vt-P2 were found to be high (5.4 x $10^6$ and 2.6 x $10^5$ genome copies per µL of total extracted DNA, respectively). Gross lesions in wt-infected placentas, wt-P1 (Fig 1B) and wt-P2 (S3 Fig), affected approximately 60% of the placental surface, showing dark red cotyledons and thickened red intercotyledonary membranes, with a grey or whitish exudate adherent on the surface. These wt lesions were indistinguishable from those observed in the vt placentas.

## PCR-RFLP analysis of qPCR-positive placental samples

PCR-RFLP analysis of the isolated organism and the two qPCR-positive placental samples (vt-P1 and -P2) revealed restriction enzyme digestion patterns identical to those obtained for the *C. abortus* 1B vaccine strain rather than those obtained for wt strain S26/3 or wt vaccine parent strain AB7 (Fig 2). Specifically, we observed no cleavage of the 319 bp CAB153 and 118 bp CAB636 PCR fragments with restriction enzymes *SfcI* and *HaeIII*, respectively, with genomic DNA from either of the placental samples originating from ewe 6093. This contrast with the control wt samples where cleavage occurred. Similarly, the 177 bp CAB648 (and equivalent genes) PCR products derived from the two ewe 6093 placental samples (vt-P1 and -P2) were only cleaved once with *Sau3AI*, producing fragments of 89 and 88 bp in size, as also observed for the 1B vaccine strain. In contrast, for the wt strains the 88 bp fragment was cleaved further producing fragments of 67 and 21 bp. There was no evidence of mixed vt and wt PCR products present in either of the placental samples from ewe 6093.

## Placental histopathology

The placentas from ewe 6093 (vt-P1 and -P2) that were *C. abortus* qPCR-positive and PCR-RFLP vt-positive exhibited pathology typical of that observed in EAE following a wt *C. abortus* infection. Placentas vt-P1 and vt-P2 showed lesions indistinguishable from both wt placentas (wt-P1 and -P2). In vt-P1, histological lesions were multifocal to diffuse, suppurative and necrotising (Fig 3A–3C) in the majority of the cotyledons (43/48). These lesions were associated with degenerative to necrotising vasculitis and occasional fibrinoid thrombosis (Fig 4A–4C). The chorionic epithelium (trophoblast layer) was severely affected, with extensive areas of necrosis, commonly associated with a suppurative inflammatory infiltrate. In affected areas, the trophoblast cells displayed severe disruption with abundant necrotic cellular debris, occasional foci of mineralisation, presence of intracytoplasmatic inclusion bodies (Fig 5A and

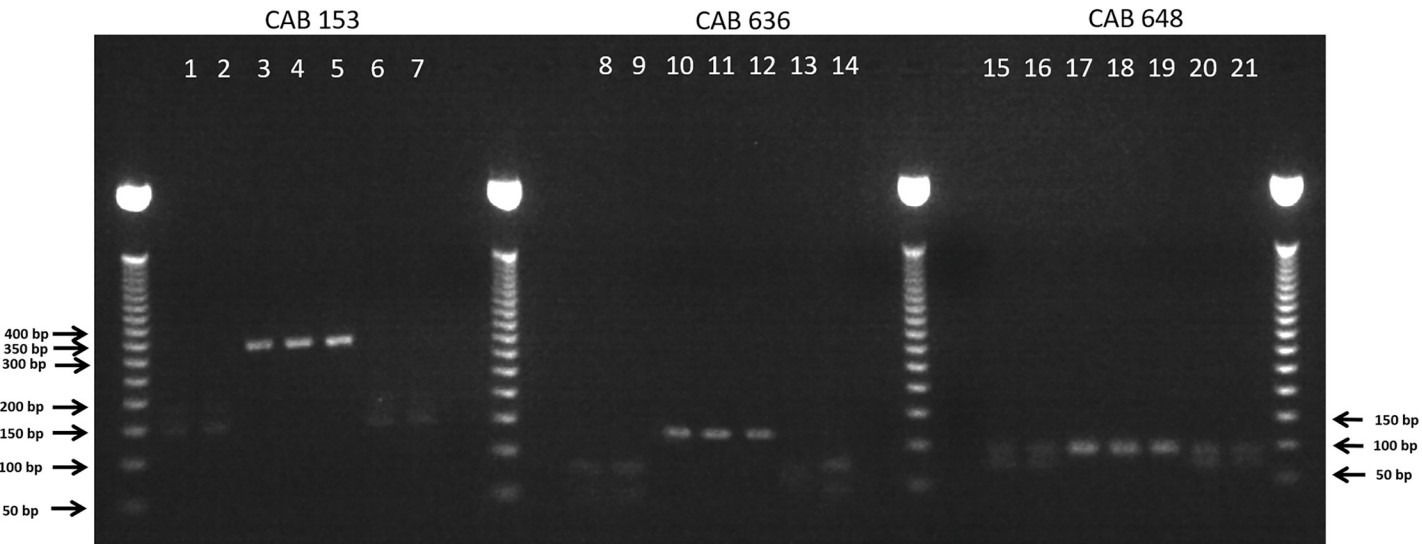

**Fig 2. PCR-RFLP analysis of *C. abortus*-positive placental samples.** Restriction enzyme digestion patterns for vt-P1 (lanes 4, 11 and 18) and vt-P2 (lanes 5, 12 and 19), wt-P1 (6, 13 and 20) and wt-P2 (7, 14 and 21), control wt *C. abortus* strains S26/3 (1,8 and 15) and AB7 (2, 9 and 16) and control *C. abortus* 1B vaccine strain (3, 10 and 17) of PCR fragments (see Materials and methods and Table 1) with enzymes *Sfc*I (lanes 1–7), *Hae*III (lanes 8–14) and *Sau3*AI (lanes 15–21).

5C) and mixed inflammatory infiltration, chiefly comprising polymorphonuclear neutrophils (PMN). Immediately under the necrotic areas of the trophoblast layer, the basement membrane and lamina propria showed a densely packed layer of mixed inflammatory cells, predominantly PMN. In affected cotyledons, multifocal vasculitis was also present. These foci were characterized by a mixed inflammatory infiltration of mononuclear cells morphologically resembling macrophages and lymphocytes and a small number of PMN in the tunica media and tunica adventitia, associated with mural necrosis and occasional occlusive fibrinoid thrombosis. Luminal occlusions in arteries and arterioles associated with the thrombosis was variable in terms of severity. These occlusions were accompanied by areas of ischemic necrosis, presenting as extensive areas of fibrinoid material, which consisted of cellular debris, karyorrhectic cells and degenerated PMN (Figs 3B, 3C and 4B). In the chorioallantoic mesenchyme, severe oedema was observed, with a mixed inflammatory infiltration composed mainly by macrophages-like cells, lymphocytic cells, and a small number of PMN (Fig 3B). In this area, occasional necrotizing vasculitis was also observed, with fibrinoid mural necrosis, and mixed inflammatory infiltration and lytic cellular debris in the lamina media and adventitia. Fibrinoid thrombosis in arteries and arterioles was variable in terms of luminal occlusion (Fig 4A–4C).

Lesions in placenta vt-P2 were also indistinguishable from wt infected placentas, showing severe, multifocal, suppurative, necrotising, placentitis (Fig 3D–3F) in most of the cotyledons (43/52) associated with necrotizing vasculitis and thrombosis (Fig 4D). In the trophoblast cells of the cotyledons, multifocal necrosis was observed mainly associated with a severe infiltration of PMN and fewer lymphocytic cells (Fig 3F). The chorioallantoic mesenchyme showed interstitial oedema with a mixed, multifocal, moderate inflammatory infiltration, composed principally of macrophage-like cells and scarce PMN and lymphocytic cells. In blood vessels, arterioles showed mixed cell infiltration of PMN and scarce lymphocytic cells, fibrinoid necrosis of the tunica media and with occasional thrombosis resulting in variable degrees of luminal occlusion (Fig 4D).

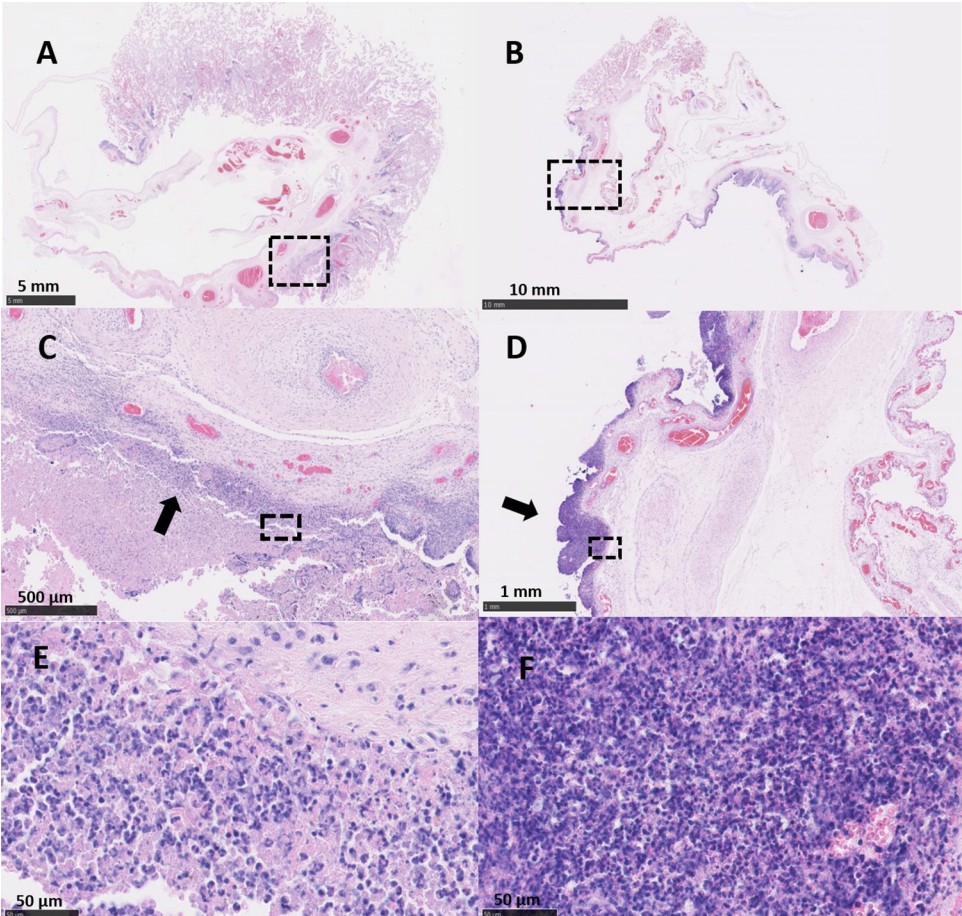

**Fig 3. Histopathological changes in the placentas of sheep infected with *C. abortus* vaccine-type strains.** (A, C and E) Placenta vt-P1: showing (A) necrosuppurative placentitis and (C) inflammatory infiltration forming a compact layer consisting primarily of large numbers of leucocytes in the epithelium of the cotyledon (arrow) attached to superficial amorphous necrosuppurative material; and (E) mixed infiltration, primarily of PMN, in different stages of degeneration, and abundant necrotic debris. (B, D and F) Placenta vt-P2: showing (B) necrosuppurative placentitis and (D) extensive compact layer of leucocytes in the trophoblast layer and basement membrane (arrow); and (F) a heavy and compact suppurative infiltration. Black outlined areas shown in top images indicates areas expanded in images located immediately below. (Scale bar: A: 5 mm; B: 10 mm; C: 500μm; D: 1 mm; E, F: 50μm).

Histopathologically, the lesions in both wt infected placentas (wt-P1 and -P2) were similar to the ones observed in the vt-placentas. The wt-placentas displayed multifocal to diffuse, suppurative and necrotising placentitis (29/33 cotyledons in wt-P1, and 38/38 in wt-P2) (Fig 6A–6F) with foci of mineralization (Fig 6D), thrombosis and degenerative to necrotizing vasculitis (Fig 4E and 4F). Extended areas of trophoblast cells exhibited abundant necrotic material, intracytoplasmic inclusion bodies (Fig 5E and 5G) and intense leucocyte infiltration, primarily PMN, forming a dense band of leucocytes in the basement membrane (Fig 6A–6D). Chorioallantois mesenchyme showed interstitial oedema, foci of haemorrhage and multifocal mixed inflammatory infiltrate composed of macrophage-like and lymphocytic cells, and PMN (Fig 6C–6F). In these areas, affected arterioles and arteries showed marked perivascular mixed inflammatory infiltration, occlusive thrombosis, and fibrinoid mural necrosis affecting the tunica intima and media of the vessels (Fig 4E and 4F). As noted in the vt-placentas, the degree of the luminal occlusion of the blood vessels was variable and was associated with extensive

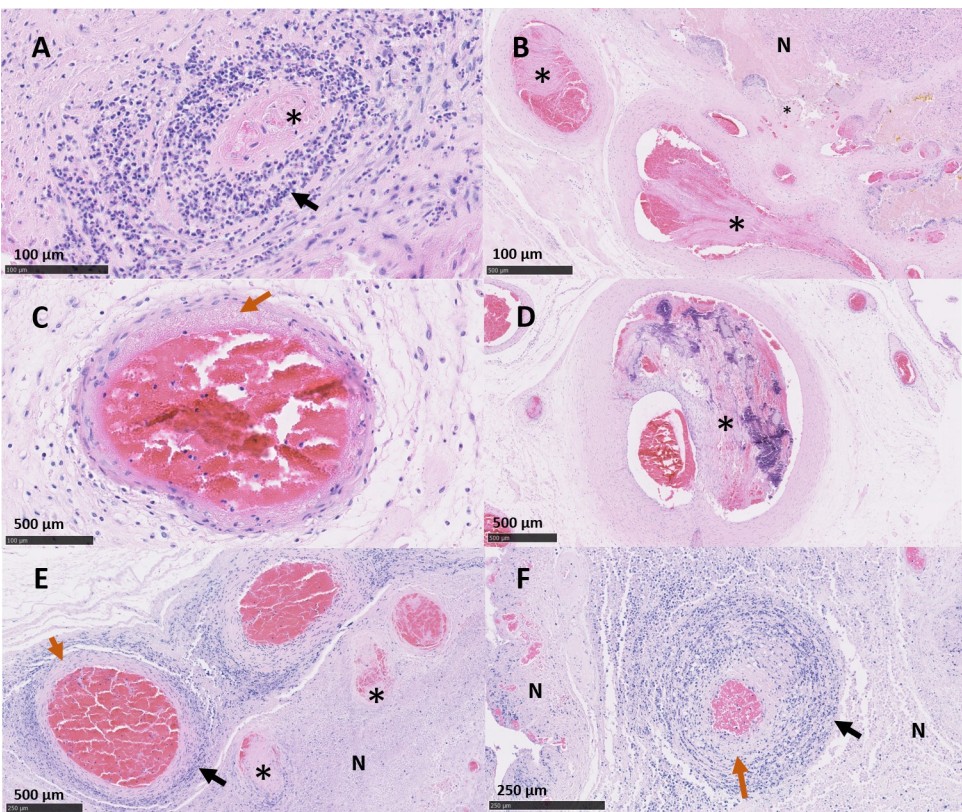

**Fig 4. Histopathological changes in the placental blood vessels of sheep infected with *C. abortus* vaccine-type and wild-type strains.** (A-C) Placenta vt-P1: showing (A) necrosuppurative vasculitis, with severe mixed inflammatory infiltration (black arrow) and partially occlusive thrombosis (*); (B) partially occlusive fibrinoid thrombosis in multiple vessels (*) with revascularisation, and ischemic necrosis of the trophoblast cells (N); (C) severe fibrinoid necrotising arteritis displaying a dense band of amorphous and intensely eosinophilic material (orange arrow); (D) Placenta vt-P2: showing severe occlusive thrombosis, with neovascularization, mineralization of the thrombi (*). (E) Placenta wt-P1: showing severe fibrinoid necrotising mural arteritis (orange arrow), multiple foci of thrombosis (*), severe perivascular inflammatory infiltration (black arrow) and ischemic necrosis of the trophoblast cells (N). (F) Placenta wt-P2, severe necrotising arteritis, showing mural degeneration (orange arrow), mixed infiltration (black arrow) and ischemic necrosis of the surrounding tissue (N). Haematoxylin and eosin. (Scale bar: A, B: 100μm; C, D, E: 500μm; F: 250μm,).

areas of necrosis in the trophoblast cells (Figs 4E, 4F, 6C and 6D). No EAE-associated lesions were observed in any of the negative control placentas (Neg-P1 and Neg-P2) (S4A–S4D Fig).

## Placental immunohistochemistry

Immunohistochemical labelling for *C. abortus* using the genus-specific anti-LPS mAb 13/4 in the vt infected placentas revealed intense positivity in most cotyledons. Only five (5/48*)* cotyledons in the vt-P1 and three (3/52) in the vt-P2 appeared negative. In both vt placentas, the strongest labelling was found mainly in chorionic trophoblast cells with intracytoplasmic inclusion bodies being evident (Fig 5B and 5D) and with a distribution that was principally multifocal to coalescing (Fig 7A and 7D). In the chorioallantoic mesenchyme, the positivity was similar to that in the trophoblast layer, but the distribution was predominantly focal. Similar labelling was observed in the wt-P1 and wt-P2 placentas, again with intracytoplasmic inlcusions being detected (Fig 5F and 5H), and only one of the cotyledons from placenta wt-P1 was negative. Both wt placentas also showed strong positive mainly multifocal to coalescing labelling in the trophoblast layer (Fig 7E and 7G) and moderate to intense multifocal labelling in

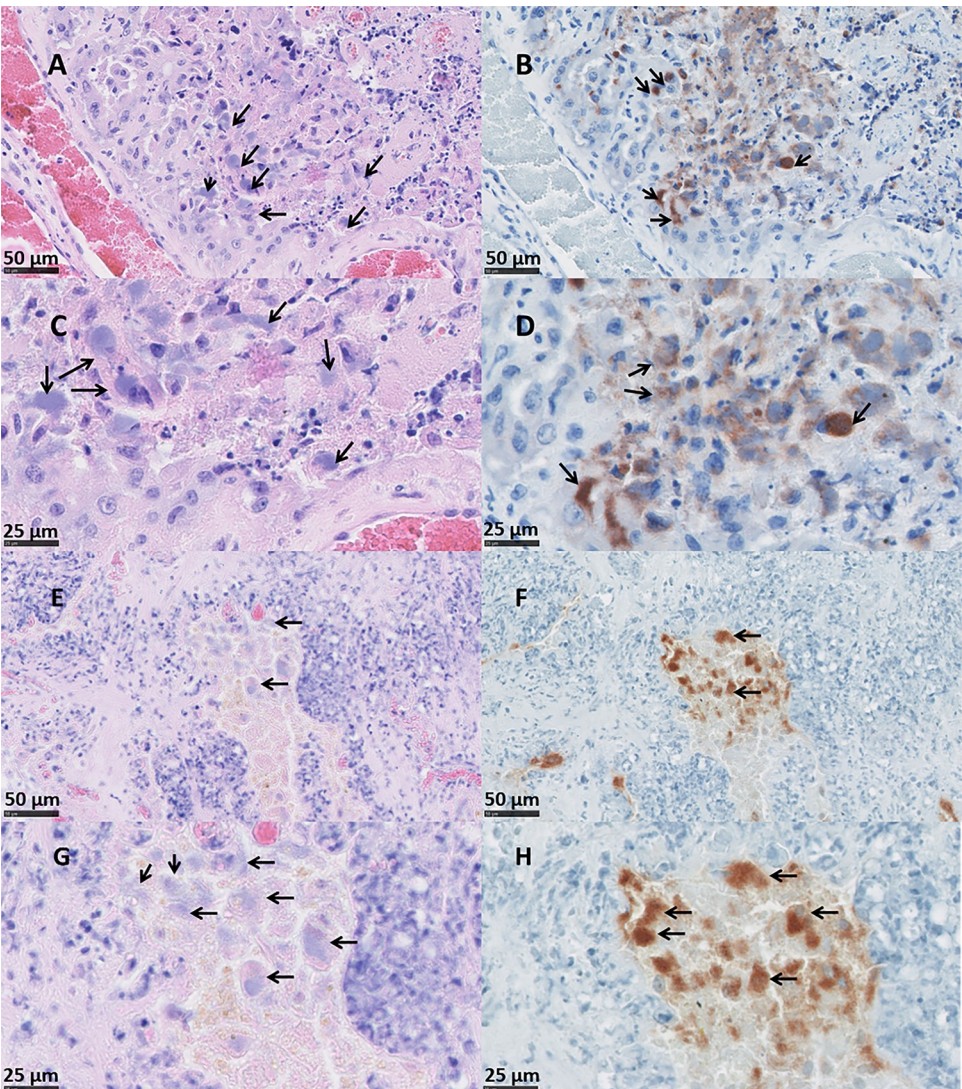

**Fig 5. Trophoblast cells showing intracytoplasmic inclusion bodies of *C. abortus* vt- and wt-strains in sheep placentas.** Placentas vt-P1 (A-D) and wt-P1 (E-H) were stained by HE for histological examination (A and C for vt-P1; E and G for wt-P1) and chlamydial organisms labelled by IHC using genus-specific anti-LPS mAb 13/4 and counterstained with haemotoxylin (B and D for vt-P1; F and H for wt-P1). Note the presence of cytoplasmic chlamydial inclusion bodies in trophoblast cells indicated by arrows. (Scale bar: A, B, E, F: 50 µm; C, D, G, H: 25 µm).

the mesenchyme. In all the placentas evaluated, vt and wt, blood vessels were negative in most of the tissues, except for 4 arterioles (one in vt-P2, one in wt-P1 and two in wt-P2), that showed focal positive labelling in the tunica intima of the arterioles. No positive IHC was observed in the cotyledons from the negative control placentas (Neg-P1 and Neg-P2) (S4E and S4H Fig).

## Discussion

In this study, we examined placentas from a flock of sheep that had been vaccinated twice with the 1B vaccine strain over different breeding seasons, to look for evidence of pathology associated with the presence of the vaccine strain. No abortions occurred in this flock, with all ewes reported as delivering 'apparently normal healthy' live lambs. It was a limitation of the study

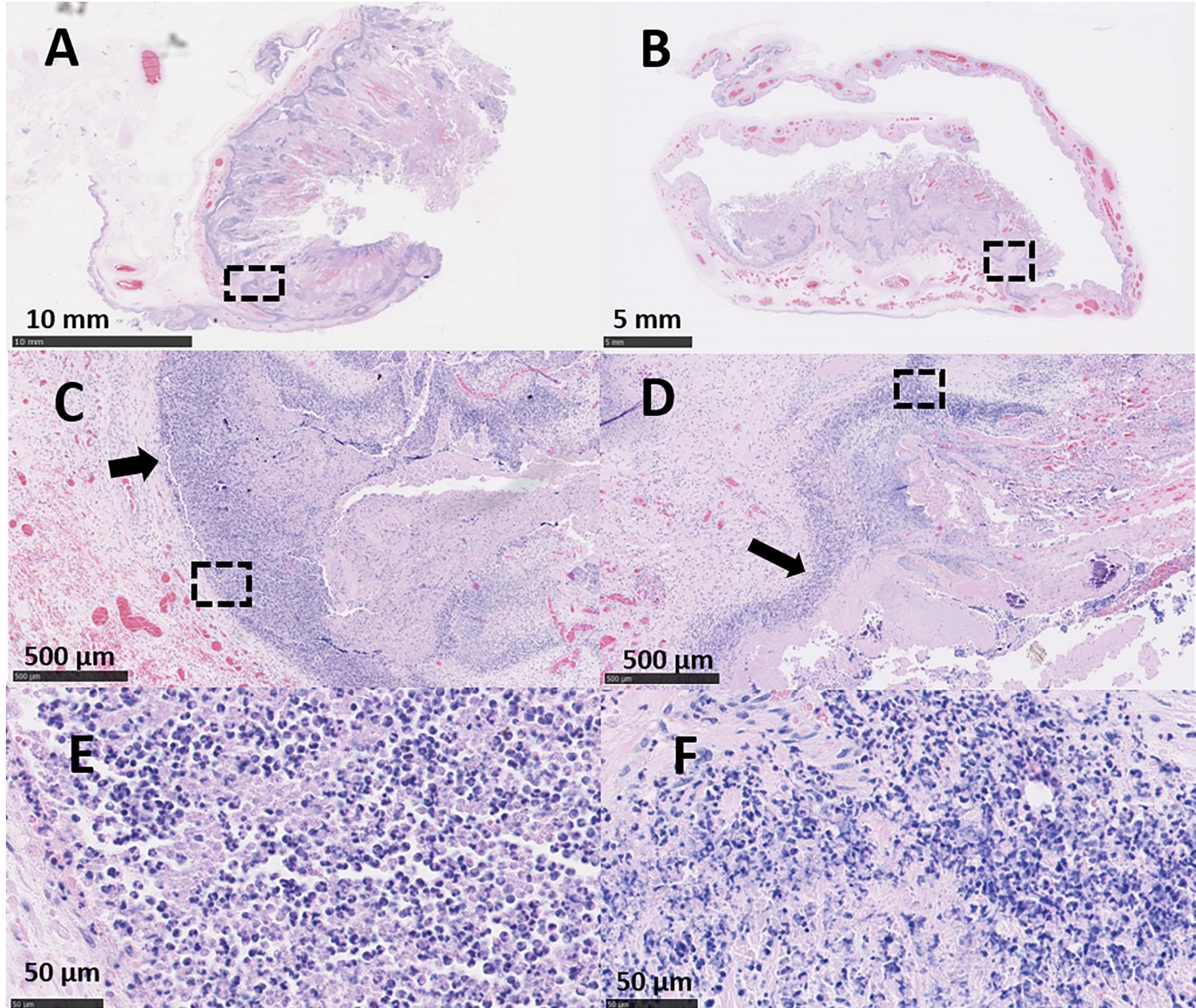

**Fig 6. Histopathological changes in the placentas of sheep infected with *C. abortus* wild-type strains.** (A, C and E) Placenta wt-P1: note (A) necrosuppurative placentitis, (C) extensive, severe, suppurative infiltration of the trophoblast cells, necrotic material (N) in the luminal face of the layer and multiple haemorrhages in the basal tissue. (E) wt-P1: suppurative infiltration and necrotic debris in the trophoblast cells; (B, D and F) Placenta wt-P2, (B) necrosuppurative placentitis (D) fibrinoid necrosis of the trophoblast cells (N) with foci of mineralization (orange arrow) and suppurative infiltration in forming a dense layer of leucocytes in the basement membrane (black arrow), (F) suppurative infiltration containing PMN in different degrees of degeneration and necrotic debris. Black outlined areas in images represents areas expanded in images located in their immediate below (Scale bar: A: 10 mm; B: 5 mm; C, D: 500μm; E, F: 50μm).

that we were unable to follow the lambs during the neonatal period to follow up their clinical status and ensure that they were free of *C. abortus* infection. From the initial assessment of the placentas we identified one that exhibited gross pathology which was characteristic of the pathology typically observed in cases of EAE. The presence of chlamydial organisms was confirmed in this case by mZN staining of a placental smear. However, smears of all of the other collected placentas revealed another placenta to be also positive. As all sampling was undertaken blind it was only later that we determined that this additional placenta was from the

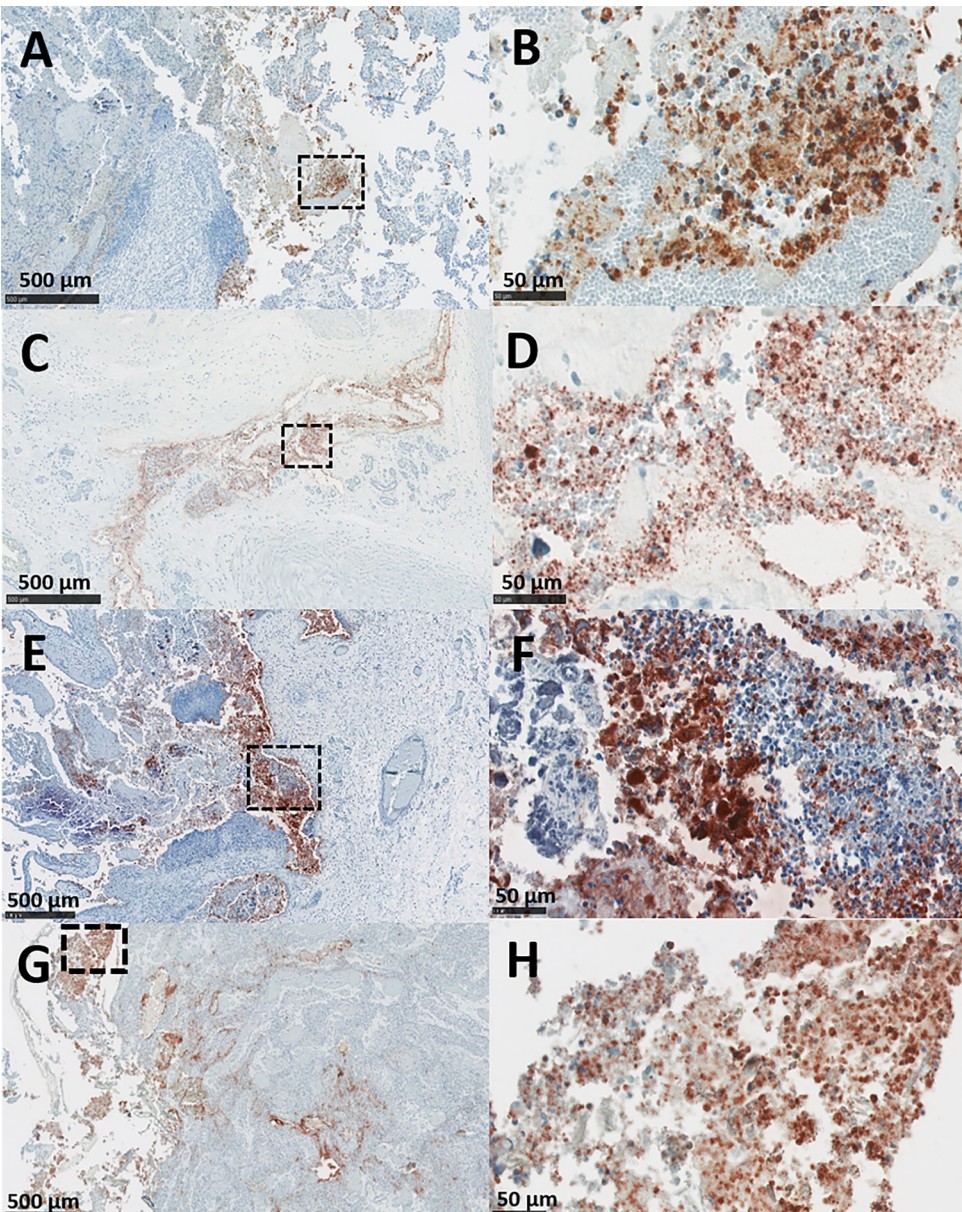

**Fig 7. Immunohistochemical detection of chlamydial antigen in ovine placentas.** Both vt- and wt-placentas were labelled by IHC using genus-specific anti-LPS mAb 13/4 and counterstained with haematoxylin. (A and B) Placenta vt-P1: showing (A) diffuse positive labelling in the cotyledonary trophoblast layer, with (B) intense and abundant labelling of the trophoblastic cells. (C and D) Placenta vt-P2: showing (C) positive labelling of the trophoblast layer, with (D) the intense multifocal to coalescent labelling of the trophoblast cells. (E and F) Placenta wt-P1: showing (E) intense multifocal to coalescent immunolabelling (F) with suppurative infiltration in the trophoblast layer. (G and H) Placenta wt-P2: diffuse moderate labelling of the trophoblast cells, showing (H) intense labelling of the individual mononuclear cell surrounding by abundant cellular debris. The outlined black squares in the images on the left indicate the magnified area shown in the images on their immediate right. IHC anti-chlamydia mAb 13/4. (Scale bar. A, C, E and G: 500µm; B, D, F, and H: 50µm).

same ewe. It is not unusual for the extent of gross pathology to vary in placentas from the same ewe and in different ewes that abort all lambs, or deliver a combination of dead, weak or 'apparently normal' lambs [1]. Generally, the more gross pathological damage that is observed

(typically 60–100% of the placental surface) the more likely that an abortion will occur (due to placental insufficiency resulting in a lack of nutrients and oxygenated blood getting to the fetus), but conversely there are instances where a heavily infected placenta is delivered following the birth of an 'apparently normal' live lamb [6]. Indeed, in this study, although vt-P1 exhibited around 60% gross pathology, the ewe delivered what appeared to be normal healthy lambs. Thus, even though the animal delivered these healthy lambs the fact that the placenta was heavily infected and viable elementary bodies were recoverable and confirmed as vt, this represents an important source and risk for transmission of infection to naïve animals. Indeed, this risk could be considered greater than following an abortion event as it is possible that the farmer would not notice such a heavily infected placenta when the lamb was born alive and therefore would fail to send it to the local veterinary laboratory for diagnosis. Furthermore, a placenta that is less infected from a live lambing, such as vt-P2 would be even less likely to be noticed, and therefore also pose an important and significant risk factor for infection transmission. Thus, although the extent of visible gross pathology may be indicative of pregnancy outcome, this is not always the case, while delivery of live lambs alone is not a good indication that a flock is free of infection. Instead, if there is any hint of a possible chlamydial infection in a flock, a full histological examination of many of the placental cotyledons would ideally be required to properly determine the extent of infection and tissue damage. But of course, this is not routinely feasible in a diagnostic laboratory setting.

The damage to the placenta, known as necrosuppurative placentitis, is caused by a combination of the action of the bacteria as well as the ensuing immunoreactivity associated with placental infection [16,24,25]. In the present study, even though the placenta infected with the vt strain showed lesions and was positive by PCR, mZN and IHC for *C. abortus*, the ewe gave birth to two 'normal' lambs. Furthermore, although both vt placentas presented different degrees of lesions, these lesions were typical of EAE and indistinguishable from those resulting from the wt S26/3 strain. This suggests that, at least in terms of gross placental pathology, the lesions found in these vt infected placentas are very similar to those resulting from a typical wt infection. Furthermore, although there is a difference in the extent of the placenta affected, a very similar number of *C. abortus* genome copies were found by real-time qPCR within a specific pathological lesion. This is consistent with what has been observed in other studies [6], where although there has been little evidence of gross pathology we still observe high bacterial loads following qPCR analysis. Therefore, qPCR analysis of placental tissue, while not necessarily directly related to the extent of the gross pathology or the clinical outcome, is important for assessing any potential risk of transmission of infection to other naive animals.

Quantitative PCR analysis of all of the placentas confirmed that only two placentas were positive for *Chlamydia abortus*, while PCR-RFLP analysis confirmed the genotype in both cases to be vt rather than wt. There was no evidence of any mixed infection in either of the samples, as observed in other studies [11,13,14,22], suggesting that any observed gross pathology was due to the vaccine rather than a wild-type *C. abortus* strain. Furthermore, bacterial load estimated by qPCR also showed the presence of *C. abortus* DNA in cotyledons to be very similar to those observed in a normal field infection resulting from a wt strain. While there were no abortions in this flock, possibly as a result of the two vaccinations giving added protective immunity, the identification of even a single animal with infection and live organisms in the placentas could be enough to cause transmission to other naïve animals, leading to infection and possible abortion in the subsequent breeding season [3]. The fact that the vt strain identified in these placentas originated from the administered vaccine strain would then imply a possible risk of transmission of this vaccine strain to other unvaccinated naïve animals, as has been recently reported to occur in flocks in France [11].

Histologically, EAE typically shows a multifocal to diffuse, suppurative, necrotising placentitis resulting from infection of the placental chorionic epithelium and caused by a combination of the action of the bacteria as well as the ensuing immunoreactive response. Infection is thought to occur following the transmission of chlamydial EBs from the uterine caruncle to the trophoblast cells of the chorionic epithelium in the fetal cotyledon, probably as a result of the leakage of blood from capillaries in the maternal caruncle septal tips that occurs around day 60 of gestation [10]. From here the chlamydial EBs replicate and form the readily visible cytoplasmic inclusions observed in both the wt and vt placentas in this study, as well as in other published studies [16]. These pathological changes have generally been reported as occurring after around day 84–90 of gestation [10]. As the surrounding trophoblast cells are destroyed by the invading chlamydial organisms the inclusions become less defined, with a more diffuse staining resulting from release of the EBs being mostly evident. During this time infection spreads to the peri-placentome and then to the intercotyledonary regions, resulting in epithelial damage, oedema and inflammation that gives rise to the characteristic reddened and thickened placental membranes that are associated with this disease [6,9,16,17] and which was also observed in this study. Histological assessment of the two vt positive placentas revealed a mixed inflammatory cell infiltrate, characterised mainly by PMN and a lesser number of macrophages and lymphocytic cells, with vasculitis and occasional occlusive fibrinous thrombosis. Furthermore, the lesions from these two placentas were found to be indistinguishable from each other, differing only in terms of the extent of tissue damage, as well as indistinguishable from those observed in classical wt disease, either as judged by the positive control samples used in the present study or as reported previously in other studies [10,15]. Such observed differences in placental pathology in multiple placentas from the same animal is a common observation for this disease [6,17]. Thus, overall, the results suggest that there is no difference between the lesions resulting from a normal wild-type infection and those resulting from an infection caused by administration of the commercial *C. abortus* 1B vaccine strain.

## Conclusion

This study has provided strong molecular evidence for the presence of only the *C. abortus* live attenuated 1B vaccine strain in the placentas of a ewe exhibiting gross pathology characteristic of that observed following a wt infection. Furthermore, the bacterial load was similar to that observed in such typical wt cases. Histological and immunohistochemical analysis showed no discernible difference in pathology from that observed in classical cases of EAE. Therefore, the results suggest that the vt strain can cause the typical placental lesions that are observed following a wt strain infection in ewes. These results complement those of an earlier study [17] where administration of a high dose of a virulent wt strain (same wt strain as used in this study), which is similar to the dose of the commercial vaccine that is administered, stimulated protective immunity and resulted in protection from abortion. Combined with evidence that the live vaccine strain is not attenuated [3], this suggests that the vaccine strain provides protection through the high doses that are administered rather than because of any attenuation. Although there was no abortion associated with the pathology in this study, the fact that a high bacterial load was present in the placentas has potentially important implications regarding transmission of the live vaccine to other unvaccinated naïve animals, where it could result in disease and abortion. Similar consideration should be given in terms of human health as the pathogen is zoonotic. However, we should also bear in mind that the vaccine has been used successfully for controlling the disease for many years since it introduction around 1995 [1,8,14,26,27], and therefore should continue to be used until a suitable replacement has been developed.

## Supporting information

**S1 Fig. Placentas vt-P2 from ewes infected with vt strain formalin-fixed.** Placenta vt-P2, note the difference in the coloration of the cotyledons and the thickening of the intercotyledonary membranes of the marked areas (black circles).
(TIF)

**S2 Fig. Boxplot. Genome copies of *C. abortus* per µl of placental DNA.** Boxplot showing the number of genome copies (natural logarithmic scale) of *C. abortus* per µl of extracted genomic DNA in the 117 placentas analysed by qPCR. The two outlier points correspond to vt-P1 and vt-P2.
(TIF)

**S3 Fig. Placentas from ewes infected with *C. abortus* s26/3 wild-type strains, and negative control.** (A) Placenta wt-P2 showing the oedema and thickening of the whole placenta partially covered with a cream-colored exudate on the intercotyledonary area and dark red or grey cotyledons. (B) Placenta Neg-P2 showing red colour of the cotyledons, and thin and clear intercotyledonary membranes.
(TIF)

**S4 Fig. HE and IHC in ovine placentas.** Ovine placentas infected with Neg-P1 (A, B, E and F) and Neg-P2 (C, D, G and H) strained with HE (A-D) and with IHC using genus-specific anti-LPS mAb 13/4 and counterstained with haematoxylin (E-H). (AB) Placenta Neg-P1: showing (A) placental epithelium showing typical trophoblastic cells. (C-D) Placenta Neg-P2: showing congestion of the blood vessel with absence of thrombosis and perivascular infiltration. (E-F) Placenta Neg-P1: showing (E) Negative immunolabelling of the trophoblast cells. (G-H) Placenta Neg-P2: Negative labelling of the trophoblast layer. The outlined black squares in the images on the left indicate the magnified area shown in the images on their immediate right. IHC anti-chlamydia mAb 13/4. (Scale bar. A, C, E and G: 500µm; B, D, F, and H: 50µm).
(TIF)

**S5 Fig. Raw gel image corresponding to Fig 2 "PCR-RFLP analysis of *C. abortus*-positive placental samples".**
(TIF)

## Acknowledgments

The authors would like to thank Val Forbes, Clare Underwood, Alice Plat and Moredun Bioservices for valuable technical assistance in helping with the processing of placentas.

## Author Contributions

**Conceptualization:** Elspeth Milne, Neil Donald Sargison, Francesca Chianini, David Longbottom.

**Data curation:** Sergio Gaston Caspe.

**Formal analysis:** Sergio Gaston Caspe, Javier Palarea-Albaladejo.

**Investigation:** Sergio Gaston Caspe, Morag Livingstone, David Frew, Kevin Aitchison, Sean Ranjan Wattegedera, Gary Entrican, Francesca Chianini, David Longbottom.

**Methodology:** Sergio Gaston Caspe, Morag Livingstone, Tom Nathan McNeilly.

**Supervision:** Elspeth Milne, Neil Donald Sargison, Francesca Chianini, David Longbottom.

**Validation:** Sergio Gaston Caspe, Morag Livingstone.

**Visualization:** Sergio Gaston Caspe.

**Writing – original draft:** Sergio Gaston Caspe, Francesca Chianini, David Longbottom.

**Writing – review & editing:** Sergio Gaston Caspe, Morag Livingstone, Sean Ranjan Wattegedera, Gary Entrican, Javier Palarea-Albaladejo, Tom Nathan McNeilly, Elspeth Milne, Neil Donald Sargison, Francesca Chianini, David Longbottom.

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
