## [Decision Letter · Decision Letter 0]

8 Oct 2020

PONE-D-20-27056

The 1B vaccine strain of *Chlamydia abortus* produces placental pathology indistinguishable from a wild type infection

PLOS ONE

Dear Dr. Caspe,

Thank you for submitting your manuscript to PLOS ONE. After careful consideration, we feel that it has merit but does not fully meet PLOS ONE’s publication criteria as it currently stands. Therefore, we invite you to submit a revised version of the manuscript that addresses the points raised during the review process.

Please provide a point by point response to each reveiwer's concerns with appropriate changes in the manuscript. 

We look forward to receiving your revised manuscript.

Kind regards,

Deborah Dean, M.D., M.P.H.

Academic Editor

PLOS ONE

Journal Requirements:

Reviewers' comments:

Reviewer's Responses to Questions

**Comments to the Author**

1. Is the manuscript technically sound, and do the data support the conclusions?

Reviewer #1: Yes

Reviewer #2: Partly

2. Has the statistical analysis been performed appropriately and rigorously? 

Reviewer #1: Yes

Reviewer #2: N/A

3. Have the authors made all data underlying the findings in their manuscript fully available?

Reviewer #1: Yes

Reviewer #2: Yes

4. Is the manuscript presented in an intelligible fashion and written in standard English?

Reviewer #1: Yes

Reviewer #2: Yes

5. Review Comments to the Author

Reviewer #1: This is a well written and interesting article. My principal comment is that I would like to see a little more balance in the discussion. The live vaccine has been shown in controlled and field studies to be effective and useful (Rodolakis et al., 1983; Chalmers et al., 1997). It is clear from this article and other work that the vaccine stain is not as attenuated as one would like and the authors rightly point out the need to consider transmission to naïve ewes and possibly also to pregnant humans. However, the evidence presented here does not show or purport to show that the vaccine strain is as virulent as a standard field strain. Therefore the vaccine may still have a role in control of this disease, while we await a better commercial vaccine.

There are several minor corrections required as listed below and indicated in the attached pdf file.

General comments:

Please refer to the ‘trophoblast cells’ or the ‘trophoblast layer’ rather than referring to it as the ‘trophoblast’. There is some repetition of results in the discussion, please edit accordingly to make the discussion more concise.

Specific comments:

Line 109: ‘secondary bacterial infection’ could these infections not also be primary bacterial infections, does it matter. Please delete the word ‘secondary’.

Lines 137 – 139: Can you supply a reference for the cultivation method employed showing that the method described is as sensitive as other standard methods. It is unusual not to include antimicrobials and an antimetabolite such as cycloheximide when cultivating chlamydiae.

Lines 144 to 164: The section “Placenta sampling….’ should come before ‘Bacteriological isolation’.

Lines 148 -151: Please clarify if the mixing of the positive cotyledon with negative cotyledons occurred before DNA extraction or after.

Line 166: What are the ‘isolated organisms’ referred to here ?, details please.

Lines 423 – 426: Please supply a primary source of the information stated here about the temperature that the vaccine strain 1B grows at (ref 3 is not suitable). The best source I could find was Rodolakis (1983, Infection and Immunity, 42:525-530) who stated that the optimum temperature for strain 1B in McCoy cells was 38ºC.

Lines 438-439: Need to re-word. It is a limitation of the study that you were unable to follow the lambs during the neonatal period. As this sentence is written it suggests that you missed other ill effects of the vaccine strain – you do not have the evidence to say this or even to imply it.

Line 452: Please supply a reference for this statement about “birth of an ‘apparently normal’ live lamb”.

Lines 460 – 462: Delete, this is repetition.

Lines 480 to 482: This needs re-wording. It is a very significant and generalised conclusion without sufficient data to back it up. Needs to be a lot more circumspect.

Line 489: qPCR will never detect the presence of antigen, please correct.

Lines 504-505: please supply a photo with arrows indicating these intracytoplasmic inclusions.

Lines 533 - 536: Some further discussion of the possible reasons/factors why the commercial vaccine has controlled EAE (despite its limitations) in many affected flocks is desirable.

Reviewer #2: In this study, Sergio et al. investigated the demonstrate that the 1B strain of Chlamydia abortus can infect the placenta, and produced typical EAE placental lesions that are indistinguishable from those found in wt infected animals. The manuscript is well written. Overall, the conclusion is supported by the performed assays.

One of the main tools used in this study was PCR-RFLP used to discriminate the 1B strain from the wild-type isolate (S26/3). While PCR-RFLP was used often to differentiate similar targets, the authors are highly encouraged to provide the DNA sequencing data (PCR products) from these two 1B samples of your work, from the vaccine, and from the S26/3 controls. DNA sequencing is so convenient and powerful nowadays, and it should be very easy for the authors to provide the DNA sequencing data of your work, and then compare these nucleotide sequences from your work with those in the GenBank. This reviewer is not sure which regions are the target of the PCR/ PCR-RFLP used in this study. The BLASTn from NCBI showed that there are 99.83% similarity between 1B and S26/3 in their membrane protein D gene (similarity: 4580/4588; zero gap).

In this study, placentas were collected from 75 ewes. The ewes looked healthy, but it remained unknown if the ewes carried Chlamydia abortus or other pathogens. After the vaccination of 1B vaccine, did abortion happen to any of those vaccinated ewes? Your Abstract reads: "...provided strong evidence that the 1B strain is not attenuated and can infect the placenta causing disease in some ewes..." You need more evidence to support this statement. In this work, 2/173 placentas were found positive for 1B (clinical abortion happened?). Experimental infection with C. abortus might resulted in 50% of abortion? The Abstract and even the title needs to modified to reflect the findings supported by your study.

6. PLOS authors have the option to publish the peer review history of their article (what does this mean?). If published, this will include your full peer review and any attached files.

Reviewer #1: No

Reviewer #2: No

---

## [Author Response · Author response to Decision Letter 0]

21 Oct 2020

Response to Reviewers

Our responses to the Reviewer’s comments are detailed below. All line numbers refer to the ‘Revised Manuscript with Track Changes’. 

Review #1 Comments to the Author:

Reviewer #1: This is a well written and interesting article. My principal comment is that I would like to see a little more balance in the discussion. The live vaccine has been shown in controlled and field studies to be effective and useful (Rodolakis et al., 1983; Chalmers et al., 1997). It is clear from this article and other work that the vaccine stain is not as attenuated as one would like and the authors rightly point out the need to consider transmission to naïve ewes and possibly also to pregnant humans. However, the evidence presented here does not show or purport to show that the vaccine strain is as virulent as a standard field strain. Therefore the vaccine may still have a role in control of this disease, while we await a better commercial vaccine.

Authors Response: We thank reviewer #1 for their kind remarks regarding the writing of the article. We note the comments regarding the balance of the Discussion. We agree that the live vaccine has successfully controlled disease outbreaks since its first use in the mid-1990s and therefore still has a place in the control of the disease. We have been very clear about this in our previous studies (Wheelhouse et al, 2010; Longbottom et al, 2013, 2018). Similarly, we have previously shown that there is no genetic basis for any attenuation in the vaccine strain (Longbottom et al, 2018), which supports our previous working hypothesis on the way the vaccine works, as described in Longbottom et al (2013). We did not feel it necessary to repeat these points in this study in any great detail, which is purely aimed at describing the pathology resulting from an infection due to the vaccine strain rather than addressing any of these specific points. However, we did make a comment in the Conclusion to briefly discuss this point. In light of the comments of this reviewer we have modified the Discussion in places and the Conclusion to make these points clear and hopefully address the balance issue raised by reviewer. This has also resulted in the inclusion of the two references that the reviewer mentions (refs #26 and 27). See lines 581-583 and 590-592. 

Reviewer #1: There are several minor corrections required as listed below and indicated in the attached pdf file.

General comments:

Please refer to the ‘trophoblast cells’ or the ‘trophoblast layer’ rather than referring to it as the ‘trophoblast’. There is some repetition of results in the discussion, please edit accordingly to make the discussion more concise.

Authors Response: We have made the corrections regarding ‘trophoblast cells/layer’ throughout the manuscript. Similarly, we have addressed the minor edits raised by the reviewer in the pdf file provided (line 88, 92, 147-152, 170-171, 211, 227, 264, 325, 333, 338, 346, 395, 402, 403, 407, 444, 493-494, 500, 508, 512-514, 517-519, 528-529, 539). We have read through the Discussion again regarding making it more concise by removing repetition of results – we had left some of this in to make it easier for the reader to follow. We have also removed the first paragraph in the Discussion, which is largely repetition of the Introduction, to make it more concise, as requested by the reviewer.

Reviewer #1: Specific comments:

Line 109: ‘secondary bacterial infection’ could these infections not also be primary bacterial infections, does it matter. Please delete the word ‘secondary’.

Authors Response: The reviewer is quite correct, this was a mistake and the word ‘secondary’ has now been removed (line 111). 

Reviewer #1: Lines 137 – 139: Can you supply a reference for the cultivation method employed showing that the method described is as sensitive as other standard methods. It is unusual not to include antimicrobials and an antimetabolite such as cycloheximide when cultivating chlamydiae.

Authors Response: We thank the reviewer for pointing this out. This was an omission on our part and is a routine standard method employing the use of antimicrobials and antimetabolite cycloheximide. This has now been rectified in the revised version (lines 177-179). 

Reviewer #1: Lines 144 to 164: The section “Placenta sampling….’ should come before ‘Bacteriological isolation’.

Authors Response: Looking at the structure of the M&Ms again we agree with the reviewer and have moved ‘Bacteriological Isolation’ after ‘Placenta sampling…’.

Reviewer #1: Lines 148 -151: Please clarify if the mixing of the positive cotyledon with negative cotyledons occurred before DNA extraction or after.

Authors Response: We have already stated that mixing occurred before DNA extraction in the original text “following extraction of DNA from the pooled tissue” (line 153).

Reviewer #1: Line 166: What are the ‘isolated organisms’ referred to here ?, details please.

Authors Response: This refers to the ‘Bacteriological Isolation’ section that has been moved above, and are the organisms isolated as a result of that. We have inserted this in brackets to make this clearer (line 186). 

Reviewer #1: Lines 423 – 426: Please supply a primary source of the information stated here about the temperature that the vaccine strain 1B grows at (ref 3 is not suitable). The best source I could find was Rodolakis (1983, Infection and Immunity, 42:525-530) who stated that the optimum temperature for strain 1B in McCoy cells was 38ºC.

Authors Response: The reviewer is correct. However, in view of this reviewer’s other comments about repetition and being more concise in the Discussion, we have now removed the first paragraph in the Discussion (lines 459-470), which is essentially repetition from the Introduction. Therefore, no additional references need adding here, although this change has resulted in the deletion of ref #24, which in turn means a renumbering of those that follow.

Reviewer #1: Lines 438-439: Need to re-word. It is a limitation of the study that you were unable to follow the lambs during the neonatal period. As this sentence is written it suggests that you missed other ill effects of the vaccine strain – you do not have the evidence to say this or even to imply it.

Authors Response: We understand the reviewer’s comment but do not completely agree with it. Yes this was a limitation of the study but nonetheless it is possible that the lambs did have an underlying health issue that we were not made aware of and feel this is worth pointing out. Even apparently normal healthy lambs may still be infected. We are not implying that they were, we are merely stating that it was a possibility that should not be ignored. We have adapted the text somewhat to tone this down and address the reviewer’s concerns (lines 475-477).

Reviewer #1: Line 452: Please supply a reference for this statement about “birth of an ‘apparently normal’ live lamb”.

Authors Response: We have added a reference as requested (ref #6) (line 491).

Reviewer #1: Lines 460 – 462: Delete, this is repetition.

Authors Response: The reviewer does not indicate why they think this sentence is repetition. We have carefully reread what we have written and this is not repetition of the text above, accordingly we wish to retain the sentence. 

Reviewer #1: Lines 480 to 482: This needs re-wording. It is a very significant and generalised conclusion without sufficient data to back it up. Needs to be a lot more circumspect.

Authors Response: We are not clear why the reviewer has made this comment with regards to this sentence. We know from other studies (eg. we have cited one of our own #6) that bacterial load does not equate to the extent of pathology. This is known to be true. Therefore, in terms of diagnosis the selection of the area to be sampled for diagnosis becomes very important as a consequence. In addition, the presence of high bacterial loads in placentas not exhibiting much gross pathology has implications for transmission. Perhaps on reflection the wording could be improved. Therefore, we have replaced this text to make this point hopefully clearer. See lines 519-524.

Reviewer #1: Line 489: qPCR will never detect the presence of antigen, please correct.

Authors Response: This has been corrected (line 534).

Reviewer #1: Lines 504-505: please supply a photo with arrows indicating these intracytoplasmic inclusions.

Authors Response: The reviewer has requested a photo to be included of the cytoplasmic inclusions within the Discussion. We feel that this would be unusual to do this in the Discussion. However, this comment made us realise that we had neglected to make this point clear in the results. Accordingly, we have added this point to the two sections “Placental histopathology” and “Placental immunohistochemistry” mentioning these cytoplasmic inclusions (lines 329-330, 399, 430 and 433-434) and then cross referencing to the new Figure (New Fig 5; Fig legend: lines 376-382). This means that we have had to renumber all subsequent figures (original figures 5-6, which now become figures 6-7) in the text.

Reviewer #1: Lines 533 - 536: Some further discussion of the possible reasons/factors why the commercial vaccine has controlled EAE (despite its limitations) in many affected flocks is desirable.

Authors Response: We referenced our previous study (ref #17) in which we discussed this point in detail and therefore did not think it appropriate to repeat in full what is discussed as a conclusion of that study. As we have stated in this paper we believe that a high dose of the organism is what triggers a protective immune response and results in either a much reduced abortion level or no abortions at all. From the perspective of this paper which is solely concerned with a description of the pathology associated with infection due to the vaccine strain compared to a wild-type strain, we feel that this is all that is required. However, we have made a slight change to the text to emphasise this point more and to add more balance on the positive aspects of administration of the commercial vaccines (lines 581-583 and 590-592).

Review #2 Comments to the Author:

Reviewer #2: In this study, Sergio et al. investigated the demonstrate that the 1B strain of Chlamydia abortus can infect the placenta, and produced typical EAE placental lesions that are indistinguishable from those found in wt infected animals. The manuscript is well written. Overall, the conclusion is supported by the performed assays.

Authors Response: We thank the reviewer for their comments regarding the writing.

Reviewer #2: One of the main tools used in this study was PCR-RFLP used to discriminate the 1B strain from the wild-type isolate (S26/3). While PCR-RFLP was used often to differentiate similar targets, the authors are highly encouraged to provide the DNA sequencing data (PCR products) from these two 1B samples of your work, from the vaccine, and from the S26/3 controls. DNA sequencing is so convenient and powerful nowadays, and it should be very easy for the authors to provide the DNA sequencing data of your work, and then compare these nucleotide sequences from your work with those in the GenBank. This reviewer is not sure which regions are the target of the PCR/ PCR-RFLP used in this study. The BLASTn from NCBI showed that there are 99.83% similarity between 1B and S26/3 in their membrane protein D gene (similarity: 4580/4588; zero gap).

Authors Response: The reviewer would have preferred that we obtain sequencing data for the PCR products instead of the validated PCR-RFLP molecular DIVA approach that we undertook. This would not provide any further information than what we have obtained by the method we used. The gene targets and the application of this PCR-RFLP have been well documented in the literature (eg refs 3, 11, 13, 14, 22 in the bibliography for this paper) and is thus a validated technique for differentiating the vaccine strain from a wild-type strain. The only difference in the PCR products between the vaccine-type and wild-type strains is a single nucleotide in the restriction enzyme cleavage sites for each gene target, so if there is no cleavage the strain is vaccine-type and if there is cleavage then it is wild-type. Therefore, we do not agree that sequencing is required. These SNPs are detailed in those published studies as well as in this study in Table 1, as are the designated gene IDs (Table 1 and lines 210-211 in Material and Methods, so we are unclear on why the reviewer is not sure what the targets are. We have clearly stated that the gene IDs are those for C. abortus strain S26/3 (lines 210-211). To help in the identification of the genome sequence of this strain from the NCBI database we have added the accession number (line 211) and we have also added a footnote to Table 1 to further clarify this. 

Reviewer #2: In this study, placentas were collected from 75 ewes. The ewes looked healthy, but it remained unknown if the ewes carried Chlamydia abortus or other pathogens. After the vaccination of 1B vaccine, did abortion happen to any of those vaccinated ewes? Your Abstract reads: "...provided strong evidence that the 1B strain is not attenuated and can infect the placenta causing disease in some ewes..." You need more evidence to support this statement. In this work, 2/173 placentas were found positive for 1B (clinical abortion happened?). Experimental infection with C. abortus might resulted in 50% of abortion? The Abstract and even the title needs to modified to reflect the findings supported by your study.

Authors Response: The reviewer asks if any of the vaccinated ewes aborted, the answer is no, which we stated on lines 120-121 in Materials and Methods, lines 262-263 in Results and in lines 473-475 in the Discussion.

We are somewhat perplexed by the reviewer’s statement concerning the comment in the abstract re ‘evidence that the 1B strain is not attenuated and can cause disease’ and that we need more evidence to support this statement. This is not a conclusion from this study, this is based on our previous published work. Although we thought that this was clear, perhaps our wording was not as clear as it could be and therefore we have made a minor change to the text to clarify this point (lines 27-28). 

The reviewer again asks if abortion occurred in the comment “In this work, 2/173 placentas were found positive for 1B (clinical abortion happened?)”, which we have addressed above, although it was actually 2/117 placentas. They also state “Experimental infection with C. abortus might resulted in 50% of abortion? “ which is generally correct for a normal wild-type infection. But then goes on to state “The Abstract and even the title needs to modified to reflect the findings supported by your study”. We do not agree with this last statement, as we think the reviewer is confusing pathology with disease outcome ie abortion. Infection and pathology do not always result in abortion. Therefore, we believe our title does accurately reflect the results and conclusion from the study we are reporting.

---

## [Decision Letter · Decision Letter 1]

4 Nov 2020

The 1B vaccine strain of *Chlamydia abortus* produces placental pathology indistinguishable from a wild type infection

PONE-D-20-27056R1

Dear Dr. Caspe,

We’re pleased to inform you that your manuscript has been judged scientifically suitable for publication and will be formally accepted for publication once it meets all outstanding technical requirements.

Kind regards,

Deborah Dean, M.D., M.P.H.

Academic Editor

PLOS ONE

Additional Editor Comments (optional):

Reviewers' comments:

Reviewer's Responses to Questions

**Comments to the Author**

1. If the authors have adequately addressed your comments raised in a previous round of review and you feel that this manuscript is now acceptable for publication, you may indicate that here to bypass the “Comments to the Author” section, enter your conflict of interest statement in the “Confidential to Editor” section, and submit your "Accept" recommendation.

Reviewer #1: All comments have been addressed

Reviewer #2: All comments have been addressed

2. Is the manuscript technically sound, and do the data support the conclusions?

Reviewer #1: Yes

Reviewer #2: Yes

3. Has the statistical analysis been performed appropriately and rigorously? 

Reviewer #1: Yes

Reviewer #2: N/A

4. Have the authors made all data underlying the findings in their manuscript fully available?

Reviewer #1: Yes

Reviewer #2: Yes

5. Is the manuscript presented in an intelligible fashion and written in standard English?

Reviewer #1: Yes

Reviewer #2: Yes

6. Review Comments to the Author

Reviewer #1: (No Response)

Reviewer #2: This reviewer appreciates the hard work from the authors to address the questions.

It is my opinion that DNA sequencing MUST be performed to make the conclusion solid in this study. I guess we have to agree that DNA sequencing is most powerful and definite tool to determine the DNA sequence. Also, the DNA sequencing is so cheap and rapid in considering that you have performed PCR and PCR products are available for nucleotide sequencing.

FYI: if we search under www.pubmed.gov, we see 926 hits using the keywords "Chlamydia DNA sequencing", and see only 28 article since 1996 using the keywords "Chlamydia, PCR-RFLP".

PCR-RFLP can be problematic with samples other than DNA from pure isolates. In contrast, DNA sequencing is so convenient, cheap, powerful, and provides definite conclusion.

7. PLOS authors have the option to publish the peer review history of their article (what does this mean?). If published, this will include your full peer review and any attached files.

Reviewer #1: No

Reviewer #2: No

---

## [Editor Report · Acceptance letter]

6 Nov 2020

PONE-D-20-27056R1 

The 1B vaccine strain of *Chlamydia abortus* produces placental pathology indistinguishable from a wild type infection 

Dear Dr. Caspe:

I'm pleased to inform you that your manuscript has been deemed suitable for publication in PLOS ONE. Congratulations! Your manuscript is now with our production department. 

Kind regards, 

on behalf of

Dr. Deborah Dean 

Academic Editor

PLOS ONE